# Cross-view Geo-localization with Layer-to-Layer Transformer

**Hongji Yang**[*]     **Xiufan Lu**[*]     **Yingying Zhu**[†]
College of Computer Science and Software Engineering
Shenzhen University
{yanghongji2020, luxiufan2019}@email.szu.edu.cn  zhuyy@szu.edu.cn

## Abstract

In this work, we address the problem of cross-view geo-localization, which estimates the geospatial location of a street view image by matching it with a database of geo-tagged aerial images. The cross-view matching task is extremely challenging due to drastic appearance and geometry differences across views. Unlike existing methods that predominantly fall back on CNN, here we devise a novel layer-to-layer Transformer (L2LTR) that utilizes the properties of self-attention in Transformer to model global dependencies, thus significantly decreasing visual ambiguities in cross-view geo-localization. We also exploit the positional encoding of the Transformer to help the L2LTR understand and correspond geometric configurations between ground and aerial images. Compared to state-of-the-art methods that impose strong assumptions on geometry knowledge, the L2LTR flexibly learns the positional embeddings through the training objective. It hence becomes more practical in many real-world scenarios. Although Transformer is well suited to our task, its vanilla self-attention mechanism independently interacts within image patches in each layer, which overlooks correlations between layers. Instead, this paper proposes a simple yet effective self-cross attention mechanism to improve the quality of learned representations. Self-cross attention models global dependencies between adjacent layers and creates short paths for effective information flow. As a result, the proposed self-cross attention leads to more stable training, improves the generalization ability, and prevents the learned intermediate features from being overly similar. Extensive experiments demonstrate that our L2LTR performs favorably against state-of-the-art methods on standard, fine-grained, and cross-dataset cross-view geo-localization tasks. The code is available online.[3]

## 1   Introduction

Estimating the geospatial location of a given image is of paramount importance for robot navigation [11], 3D reconstruction [12], and autonomous driving [5]. Recently, cross-view geo-localization, which aims to match query ground images with geo-tagged database aerial/satellite images, has emerged as a promising proposal to address this problem. Despite its appealing application prospect, the cross-view matching task is extremely challenging due to drastic viewpoint changes between ground and aerial images. Thus, it is critical to understand and correspond both image content (appearance and semantics) and spatial layout across views.

Towards the above goal, several recent works incorporate convolutional neural networks (CNNs) with NetVlad layers [8], capsule networks [20] or attention mechanisms [2, 16] to learn visually

---

[*]Equal contribution

[†]Corresponding author

[3]https://github.com/yanghongji2007/cross_view_localization_L2LTR

35th Conference on Neural Information Processing Systems (NeurIPS 2021).

discriminative representations. However, the locality assumption of their CNN architectures hinders their performance in complex scenarios, where visual interferences such as obstacles and transient objects (*e.g.*, cars and pedestrians) may exist. Instead, the human visual system utilizes not only *local* information but also *global* context to make more accurate predictions when visual signals are ambiguous or incomplete. Another branch of works exploits geometry prior knowledge to reduce ambiguities caused by geometric misalignments. Though promising, these methods either rely heavily on predefined orientation prior [9] or make a restrictive assumption that ground and aerial images are orientation-aligned [16]. Therefore, such a strong assumption limit the applicability of these approaches, which prompts us to seek a more flexible approach for encoding position-aware representations.

Motivated by these observations, we introduce Transformer [21], which excels in global contextual reasoning and thus can be naturally employed to reduce visual ambiguities in cross-view geo-localization. Besides, the positional encoding of the Transformer enables our network to learn position-dependent representations flexibly. Specifically, our proposed layer-to-layer Transformer (L2LTR) is built upon two independent Vision Transformer (ViT) [4] branches, which split a feature map into several sub-patches while modeling interactions between arbitrary patches. We show in the experiment that due to its context- and position-dependent natures, such a Transformer-based network is a well-suited candidate for cross-view geo-localization and shows its superiority compared to the dominant CNN-based counterparts.

We also take a deep look at self-attention map, which is an integral part of the Transformer and is independently learned in each Transformer block. Nevertheless, such an independent learning strategy overlooks correlations between layers. Specifically, relating features from adjacent layers could improve the representation ability of the network [13]. To explore cross-layer correlations, we replace self-attention with a novel self-cross attention mechanism. Simple yet effective, the proposed self-cross attention learns pairwise similarities between features of adjacent blocks rather than that of the same blocks. Such a cross-block interaction strategy eases the information flow across Transformer blocks, thus leading to more stable network optimization. Moreover, we empirically show that self-cross attention can improve the network's generalization ability and prevent Transformer layers from producing overly similar intermediate features. As a result, such a mechanism significantly improves the quality of image representation without increasing the model complexity.

The key contributions of this work are as follows.

- The L2LTR is the first model using a Transformer for cross-view geo-localization to the best of our knowledge. The globally context-aware nature of the L2LTR effectively reduces visual ambiguities in cross-view geo-localization, while the positional encoding endows the L2LTR with the notion of geometry, thus decreasing ambiguities caused by geometry misalignments. Since the position embeddings are learned without imposing a strong assumption on the position knowledge, the L2LTR has wider practical applicability than state-of-the-art models.

- We propose a novel self-cross attention mechanism, which interacts within cross-layer patches to ensure effective information flow across Transformer blocks. This simple yet effective design consistently enhances the representation and the generalization ability of the L2LTR without adding additional computational cost.

- Extensive experiments demonstrate that our L2LTR brings consistent and significant performance improvements for a wide range of cross-view matching tasks, including standard, fine-grained, and cross-dataset cross-view geo-localization. The L2LTR exhibits its superiority in learning visually discriminative and position-aware representations on all these tasks and achieves a new state-of-the-art performance.

## 2   Related Work

The key to cross-view geo-localization is to understand and correspond both image content (appearance and semantics) and spatial layout across views. To this end, existing cross-view geo-localization methods can be roughly grouped into content-based and geometry-based.

**Content-based methods** focus on learning image representations that are discriminative enough to distinguish between similar-looking images. Leveraging on the success of CNNs, Workman and

Jacobs [23] first introduce CNNs to the cross-view matching task. Later on, Hu *et al.* [8] incorporate a two-branch VGG [19] backbone network with NetVlad layers [1] to learn viewpoint-invariant representations. They also devise a weighted soft-margin triplet loss, which can speed up the network training. Sun *et al.* [20] apply the powerful ResNet [7] as backbone networks. Coupled with capsule layers [15], their proposed GeoCapsNet can model high-level semantics. To steer where to focus in images, the attention mechanism is introduced to the field of cross-view geo-localization. Cai *et al.* [2] introduce a lightweight attention module that combines spatial and channel attention mechanisms to emphasize visually salient features. They also propose a novel reweighting loss that adaptively allocates weights to triplets according to their difficulties, thus improving the quality of network training. SAFA [16] employs a multi-head spatial attention module to aggregate informative and diverse embedding maps. While promising, few of the above methods pay enough attention to the global dependencies of cross-view images, which hinders the discriminative ability of their embedded features. Different from existing methods, this work makes the first exploration to introduce Transformer [21] to cross-view geo-localization. We demonstrate the importance of considering global dependencies for reducing visual ambiguities.

**Geometry-based methods** aim to correspond geometric configurations between ground and aerial images, which helps to reduce ambiguities caused by geometry misalignments. To this end, Liu and Li [9] explicitly inject per-pixel orientation information into the network. Nevertheless, this is based on the assumption of accessibility of the ground-truth orientation, which is not always satisfied in practice. Shi *et al.* [16] employ a polar transform algorithm to warp satellite images so that aerial images are geometrically aligned with ground images. However, this method is only applicable to the ideal case where the ground images are orientation-aligned panoramas. Even though the dynamic similarity module proposed in [17] overcomes this limitation, the brute-force warping strategy of the polar transform overlooks the depth of the scene content. It results in obvious appearance distortions, hindering performance improvement. Regmi and Shah [14] attempt to tackle this problem by synthesizing the corresponding satellite image from a ground query using conditional GANs (cGANs), but the synthesized images are always granulated and lack details. In this paper, our L2LTR explicitly encodes learnable positional embeddings into the network without imposing a strong assumption. Unlike the previous works [9, 16, 17], as shown in the experiments, our L2LTR does not impose a strong assumption on the position knowledge but flexibly learns relative positional information.

## 3   Method: L2LTR

This paper proposes a novel layer-to-layer Transformer (L2LTR) architecture with self-cross attention mechanism for cross-view geo-localization. The following sections detail our problem setting, objective, the L2LTR architecture, and our proposed self-cross attention.

### 3.1   Problem Formulation and Objective

The goal of cross-view geo-localization is to localize a query ground image by matching it with a set of geo-tagged aerial images. We formulate this problem in the same way as prior works [9, 18, 16, 17].

Assume we have a training set $D = \{(g_1, a_1), ..., (g_N, a_N)\}$ containing $N$ cross-view image pairs of ground images $g$ and aerial images $a$. To simplify the problem, let each ground image $g_i$ correspond to only one ground-truth aerial image $a_i$ ($i \in \{1, 2, ..., N\}$) during the training phase. Given a cross-view image pair $(g_i, a_i)$, we infer the corresponding image representations as $(\mathbf{F}_i^g, \mathbf{F}_i^a)$. Then, for the $i^{th}$ exemplar, the weighted soft-margin triplet loss [8] $L$, which aims to bring matching pairs closer while pushing non-matching pairs far apart, can be defined as follows:

$$L = log(1 + e^{\alpha(d(\mathbf{F}_i^g, \mathbf{F}_i^a) - d(\mathbf{F}_i^g, \mathbf{F}_j^a))}) \tag{1}$$

where $j \in \{1, 2, ..., N\}$ and $j \neq i$. $\alpha$ is a hyperparameter used to speed up training convergence, and $d(\cdot, \cdot)$ denotes the $L_2$ distance.

### 3.2   Transformer for Cross-view Geo-localization

We seek to develop an L2LTR architecture that explores the global context and the positional information of cross-view images.

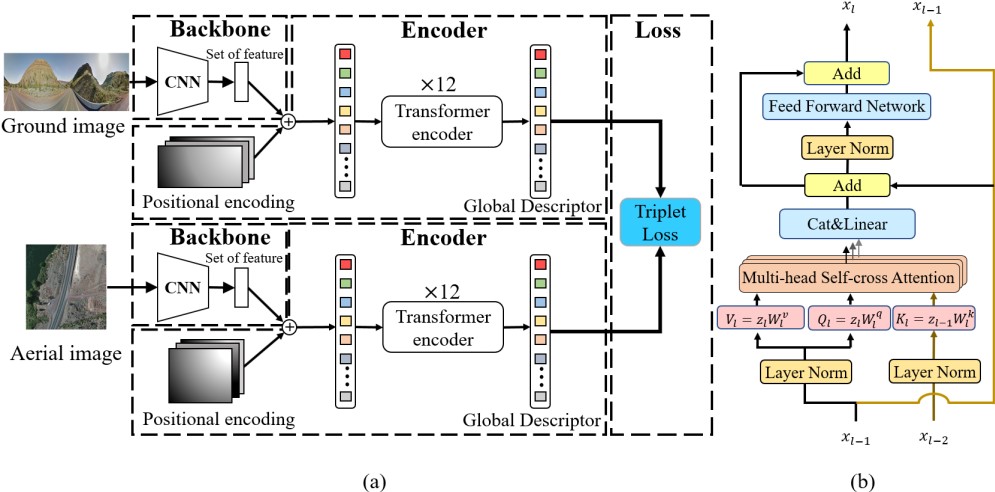

Figure 1: (a) Overview of our layer-to-layer Transformer (L2LTR). (b) Illustration of the encoder layer with self-cross attention in the L2LTR. $\mathbf{x}_{l-1}$ denotes the input of layer $l$.

**Preliminaries: Vision Transformer.** We first describe the Vision Transformer (ViT) [4] as a background. Given an image, the ViT first splits it into several patches. Then, the ViT receives as input a sequence of linear projected patch embeddings $\mathbf{x} \in \mathbb{R}^{N \times D}$, where $N$ is the number of patches, and $D$ is the patch embedding size. After prepending a learnable global representation $\mathbf{x}_{global} \in \mathbb{R}^{D}$, whose final hidden state serves as an aggregate image descriptor for the cross-view geo-localization task and adding positional embeddings $\mathbf{x}_{pos}$ to $\mathbf{x}$, we gain $\mathbf{x}_0 = [\mathbf{x}_{global}; \mathbf{x}] + \mathbf{x}_{pos}$ and feed it into an $L$-layer Transformer encoder. Each layer consists of a Multihead Self-Attention module (MSA), Feed Forward Networks (FFN), and LayerNorm blocks (LN). Note that the MSA consists of multiple self-attention heads and a linear projection block. In order to make a clear comparison with our proposed self-cross attention head, we denote the input of layer $l$ ($l \in \{1, ..., L\}$) as $\mathbf{x}_{l-1}$ and formulate a single self-attention head, the core of the vanilla MSA, as follows:

$$\mathbf{z}_l = LN(\mathbf{x}_{l-1}) \tag{2}$$

$$\mathbf{Q}_l = \mathbf{z}_l \mathbf{W}_l^q, \mathbf{K}_l = \mathbf{z}_l \mathbf{W}_l^k, \mathbf{V}_l = \mathbf{z}_l \mathbf{W}_l^v \tag{3}$$

$$\mathbf{A}_l = softmax\left(\frac{\mathbf{Q}_l \mathbf{K}_l^T}{\sqrt{D}}\right)\mathbf{V}_l \tag{4}$$

where $\mathbf{W}_l^q$, $\mathbf{W}_l^k$ and $\mathbf{W}_l^v$ are linear projection matrices.

**Domain-specific Transformer.** The drastic domain gap between the ground and aerial images makes it difficult to match cross-view images by embedding them into the same space through a domain-shared network. To suit the cross-view geo-localization task, we employ a domain-specific Siamese-like architecture with two independent branches of the same structure. Such an architecture learns ground and aerial representations separately, and it can effectively project cross-view images into a shared space when optimized with Eq. 1. The network overview is illustrated in Figure 1 (a). Each branch is a hybrid structure consisting of a ResNet backbone (denoted as "Backbone" dashed boxes in Fig. 1 (a)) extracting CNN feature map and a Transformer encoder (denoted as "Encoder" dashed boxes) modeling global context from the CNN feature map. The linear projection of patch embedding in the ViT is applied to the CNN feature map by regarding each $1 \times 1$ feature as a patch.

**Learnable positional embedding.** Incorporating geometric cues [9, 16] helps avoid ambiguities caused by geometric misalignment across views, thus greatly simplifying the cross-view geo-localization. Instead of imposing a predefined orientation knowledge on the network, this paper applies an efficient and flexible way to endow the network with the notion of geometry. Specifically, we use learnable 1D positional embeddings in the ViT, *i.e.* $\mathbf{x}_{pos} \in \mathbb{R}^{(N+1) \times D}$. By adding the positional embeddings to the patch embeddings, the transformed features become position-dependent. Furthermore, since we do not impose any assumption on position knowledge but learn it through our learning objective, our L2LTR has wider practical applicability. Experiments show that incorporating

the learnable positional embeddings helps capture relative positional information. This allows the network to geo-locate panoramas with unknown orientation and planar ground images with a limited field of view.

### 3.3 Self-cross Attention

In the vanilla ViT, the attention map is calculated independently in each layer. However, as mentioned before, such an independent learning strategy hinders the model's representation ability. To improve the quality of the learned representations, we propose a novel self-cross attention mechanism to interact features between adjacent layers and name the novel network as a layer-to-layer Transformer (L2LTR). Namely, in the L2LTR, the attention map of layer $l$ is learned not only based on $\mathbf{x}_{l-1}$ but also $\mathbf{x}_{l-2}$. Considering the semantic gap between feature maps caused by different network depths, we do not interact with all intermediate features but only the highly correlated adjacent features. Formally, in layer $l$, self-cross attention can be represented as:

$$\mathbf{z}_l = LN(\mathbf{x}_{l-1}), \mathbf{z}_{l-1} = LN(\mathbf{x}_{l-2}), \tag{5}$$

$$\mathbf{Q}_l = \mathbf{z}_l \mathbf{W}_l^q, \mathbf{K}_l = \mathbf{z}_{l-1} \mathbf{W}_l^k, \mathbf{V}_l = \mathbf{z}_l \mathbf{W}_l^v \tag{6}$$

$$\mathbf{A}_l = softmax(\frac{\mathbf{Q}_l \mathbf{K}_l^T}{\sqrt{D}})\mathbf{V}_l \tag{7}$$

Note that, for $l = 1$, we set $\mathbf{z}_{l-1} = LN(\mathbf{x}_{l-1})$. In Figure 1 (b), we illustrate the structure of self-cross attention-based encoder layer.

**How self-cross attention affects feature learning.** Compared to self-attention in Eq. 2-4, our proposed self-cross attention creates a short path between adjacent layers (highlighted in yellow in Figure 1 (b)), thus allowing information to flow effectively across layers. This shares a similar spirit with ResNet [7]. To investigate how this affects our Transformer-based network, we report the variance of recall accuracy at different training epochs on the CVUSA test set in Figure 2. Specifically, the variance curves are generated by calculating the accuracy variance over ten training epochs. Thus, a smaller variance indicates less fluctuation in recall accuracy and more stable model training. As shown, during the early training stage, the localization performance of self-attention-based model fluctuates a lot, while our L2LTR exhibits more stable performance as training processes. Furthermore, we observe that by interacting cross-layer features, self-cross attention can decrease the representation similarity between layers and enhance the network's generalization ability, thus improving the network's representation ability. This is further discussed in the experiment.

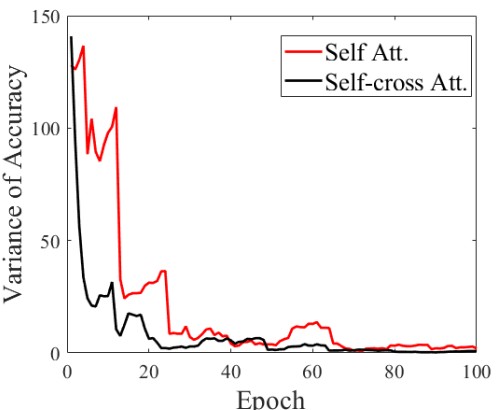

Figure 2: The variance curves of recall accuracy in the first 100 training epochs on the CVUSA test set.

## 4 Experiment

We first introduce three benchmark datasets we used to evaluate our L2LTR, evaluation protocols, and implement details of our network. Then we compare our L2LTR with state-of-the-art models in Section 4.3 and present ablation studies to illustrate the advantages of the proposed L2LTR in Section 4.4. Finally, we provide qualitative results in Section 4.5 to demonstrate the effectiveness of the positional embeddings in the L2LTR.

### 4.1 Dataset and Evaluation Protocol

**Dataset.** To verify our model's effectiveness, we conduct extensive experiments on three widely used benchmarks: CVUSA [24] and CVACT [9] (including CVACT_val and CVACT_test). The CVUSA dataset provides 35,532 image pairs for training and 8,884 image pairs for testing. The CVACT

dataset contains 35,532 pairs for training and 8,884 pairs for validation (denoted as CVACT_val). The CVACT also provides 92,802 image pairs with accurate geo-tags for testing (denoted as CVACT_test) to support fine-grained city-scale geo-localization. For the CVUSA test set and CVACT_val, the correct match of a ground image is a corresponding aerial image. For the CVACT_test, a retrieved aerial image is considered correct as long as it is within the distance d=5m from the ground-truth location of the ground image. In the experiment, we denote the tasks performed on the CVUSA test set and the CVACT_val as standard cross-view geo-localization and the tasks performed on the CVACT_test as fine-grained cross-view geo-localization.

**Evaluation protocol.** In line with [8, 9, 18, 16, 17], we evaluate our model by recall accuracy at top $K$ (r@$K$ for short, $K \in \{1, 5, 10, 1\%\}$), which represents the probability of correct match(es) ranking within the first $K$ results. For the CVUSA test set, r@1% indicates the recall accuracy at the top 1% of the test set, and for the CVACT_val and the CVACT_test, r@1% indicates the recall accuracy at the top 1% of the CVACT_val.

## 4.2 Implementation Detail

If not specified, the ground and aerial image sizes are set to $128 \times 512$ and $256 \times 256$, respectively. We empirically set model depth $L$ to 12 and initialize our L2LTR with pre-trained parameters on ImageNet [3]. The model is trained using AdamW [10] with a cosine learning rate schedule on a 32GB NVIDIA V100 GPU. The learning rate is set to 1e-4, the weight decay is chosen to 0.03, and the batch size is 32. For the weighted soft-margin triplet loss [8], $\alpha$ is set to 10.

## 4.3 Comparing L2LTR with State-of-the-art Models

Here we compare our method with several state-of-the-art methods on the CVUSA [24], CVACT_val [9], and CVACT_test [9] datasets. Unlike state-of-the-art methods that predominantly fall back on CNN, our proposed L2LTR makes the first effort to introduce Transformer to the field of cross-view geo-localization to learn globally context- and position-aware representations. Below, we verify that our L2LTR exceeds in learning visually discriminative and position-aware representations, thus achieving outstanding performance in various cross-view geo-localization tasks. Note that for a fair comparison with works [16, 17] that use polar transform [16], a kind of data pre-processing algorithm, we apply the same warping strategy to aerial

Table 1: Comparisons with state-of-the-art methods on the CVUSA [24] dataset. For all the compared methods, we cite the results from [17] and [16] if not specified. "PT" indicates whether the model applies (w/) polar transform [16] to aerial images or not (w/o).

| PT | Model | r@1 (%) | r@5 (%) | r@10 (%) | r@1% (%) |
|---|---|---|---|---|---|
| w/o | Workman *et al.* [23] | - | - | - | 34.30 |
| | Vo and Hays [22] | - | - | - | 63.70 |
| | Zhai *et al.* [24] | - | - | - | 43.20 |
| | CVM-Net [8] | 22.47 | 49.98 | 63.18 | 93.62 |
| | Liu and Li [9] | 40.79 | 66.82 | 76.36 | 96.12 |
| | Zheng *et al.* [25] | 43.91 | 66.38 | 74.58 | 91.78 |
| | Regmi and Shah [14] | 48.75 | - | 81.27 | 95.98 |
| | Siam-FCANet [2] | - | - | - | 98.30 |
| | CVFT [18] | 61.43 | 84.69 | 90.49 | 99.02 |
| | SAFA [16] | 81.15 | 94.23 | 96.85 | 99.49 |
| | L2LTR | **91.99** | **97.68** | **98.65** | **99.75** |
| w/ | SAFA [16] | 89.84 | 96.93 | 98.14 | 99.64 |
| | Shi *et al.* [17] | 91.93 | 97.50 | 98.54 | 99.67 |
| | Polar-L2LTR | **94.05** | **98.27** | **98.99** | **99.67** |

images before feeding them into the network (denoted as Polar-L2LTR) when comparing with these works. In this case, ground and warped aerial images are resized to $128 \times 512$.

**Standard cross-view geo-localization.** We first evaluate our L2LTR on standard cross-view geo-localization. Tables 1 and 2 show experimental results on the CVUSA and CVACT_val datasets, respectively. From the results, we could conclude that our L2LTR significantly surpasses the competing approaches in learning visually discriminative representations and corresponding geometric configurations across views. In particular, without applying the polar transform, our L2LTR achieves r@1 of 83.14% on the CVACT_val dataset compared to 78.28% obtained by the second-best method, while on the CVUSA dataset, the L2LTR surpasses the second-best method by a significant margin of 10.84 points at r@1. Moreover, when applying the polar transform, which geometrically aligns cross-view images, our L2LTR outperforms the competing methods, gaining 84.89% and 94.05% on the CVACT_val and CVUSA, respectively. The results indicate that the L2LTR is capable of

Table 2: Comparisons with state-of-the-art models on the CVACT_val (standard cross-view geo-localization) and CVACT_test (fine-grained geo-localization) datasets.

| PT | Model | Code Length | CVACT_val | | | | CVACT_test | | | |
|---|---|---|---|---|---|---|---|---|---|---|
| | | | r@1 (%) | r@5 (%) | r@10 (%) | r@1% (%) | r@1 (%) | r@5 (%) | r@10 (%) | r@1% (%) |
| w/o | CVM-Net [8] | 4096 | 20.15 | 45.00 | 56.87 | 87.57 | 5.41 | 14.79 | 25.63 | 54.53 |
| | Liu and Li [9] | 1536 | 46.96 | 68.28 | 75.48 | 92.01 | 19.21 | 35.97 | 43.30 | 60.69 |
| | CVFT [18] | 4096 | 61.05 | 81.33 | 86.52 | 95.93 | 26.12 | 45.33 | 53.80 | 71.69 |
| | SAFA [16] | 4096 | 78.28 | 91.60 | 93.79 | 98.15 | - | - | - | - |
| | L2LTR | **768** | **83.14** | **93.84** | **95.51** | **98.40** | **58.33** | **84.23** | **88.60** | **95.83** |
| w/ | SAFA [16] | 4096 | 81.03 | 92.80 | 94.84 | 98.17 | 55.50 | 79.94 | 85.08 | 94.49 |
| | Shi *et al.* [17] | 4096 | 82.49 | 92.44 | 93.99 | 97.32 | 35.63 | 60.07 | 69.10 | 84.75 |
| | Polar-L2LTR | **768** | **84.89** | **94.59** | **95.96** | **98.37** | **60.72** | **85.85** | **89.88** | **96.12** |

capturing visually discriminative features by modeling global context. Furthermore, we could also find that removing the polar transform algorithm leads to significant performance degradation in SAFA ($-4.21\%$ on the CVACT_val and $-8.69\%$ on the CVUSA) while less degradation is noted in our L2LTR ($-1.75\%$ on the CVACT_val and $-2.06\%$ on the CVUSA). Namely, the L2LTR does not have to rely excessively on the polar transform to establish a cross-view geometric correspondence, which could save considerable image pre-processing time on large-scale datasets. This is because adding the positional embeddings to the patch embeddings enables the L2LTR to learn position-aware representations and correspond geometric configurations between ground and aerial images. Additional qualitative evidence for this is provided in Section 4.5.

**Fine-grained cross-view geo-localization.** To evaluate the representation ability of our model, we verify the L2LTR on the fine-grained cross-view geo-localization task. Specifically, we compare the L2LTR with state-of-the-art methods on the challenging large-scale CVACT_test dataset, which is fully GPS-tagged for accurate localization. Table 2 shows the experimental results. Our L2LTR performs consistently better than all the competitors, achieving 58.33% and 60.76% at r@1 without and with the polar transform, respectively. These results further demonstrate that our L2LTR has strong representation capability.

**Cross-dataset cross-view geo-localization.** In the context of cross-view geo-localization, the transferring performance determines whether a model could be practically usable for real-life scenarios, where a query image may be dramatically different from the training ground images. As the CVACT and CVUSA datasets are collected from two different countries, they have distinctly different scene styles.

Table 3: Cross-dataset cross-view geo-localization. The results are gained by retraining and evaluating the compared models using the released codes provided by their authors.

| Model | Task | r@1 (%) | r@5 (%) | r@10 (%) | r@1% (%) |
|---|---|---|---|---|---|
| SAFA [16] | | 30.40 | 52.93 | 62.29 | 85.82 |
| Shi *et al.* [17] | CVUSA→CVACT | 33.66 | 52.17 | 59.74 | 79.67 |
| Polar-L2LTR | | **47.55** | **70.58** | **77.39** | **91.39** |
| SAFA [16] | | 21.45 | 36.55 | 43.79 | 69.83 |
| Shi *et al.* [17] | CVACT→CVUSA | 18.47 | 34.46 | 42.28 | 69.01 |
| Polar-L2LTR | | **33.00** | **51.87** | **60.63** | **84.79** |

Based on this observation, to verify the transferring performance of our model, we train the L2LTR on the CVUSA dataset and test it on the CVACT_val (denoted as CVUSA→CVACT), and vice versa. Results are reported in Table 3. We could find that our L2LTR outperforms the second-best model at r@1 by a large margin of 13.89 points on the CVUSA→CVACT task while achieving 33.00% at r@1 compared to 21.45% gained by the second-best model on the CVACT→CVUSA task. The transferring results demonstrate the outstanding generalization ability and practical applicability of our L2LTR.

**Localizing with unknown orientation and limited FoV.** As verified in Section 4.5, our L2LTR can learn relative positional information without incorporating predefined position knowledge. This makes the network generalize well to panoramas with unknown orientation and planar ground images with a limited field of view (FoV). To evaluate this point, we test the performance of our L2LTR on orientation-unknown and FoV-limited ground images. Specifically, we follow the evaluation procedure in [17], randomly shifting and cropping ground panoramas (with FoV of 360°) along the azimuthal direction on the CVUSA dataset. For a fair comparison, we set the cropped images with FoVs of 180°, 90°, and 70°. Then, we train and test the performance on ground images with the

Table 4: Comparisons with state-of-the-art models for localizing ground images with unknown orientation and limited field of view on the CVUSA.

| Train & Test FoV | 180° | | | | 90° | | | | 70° | | | |
|---|---|---|---|---|---|---|---|---|---|---|---|---|
| Model | r@1 (%) | r@5 (%) | r@10 (%) | r@1% (%) | r@1 (%) | r@5 (%) | r@10 (%) | r@1% (%) | r@1 (%) | r@5 (%) | r@10 (%) | r@1% (%) |
| CVM-Net [8] | 7.38 | 22.51 | 32.63 | 75.38 | 2.76 | 10.11 | 16.74 | 55.49 | 2.62 | 9.30 | 15.06 | 21.77 |
| CVFT [18] | 8.10 | 24.25 | 34.47 | 75.15 | 4.80 | 14.84 | 23.18 | 61.23 | 3.79 | 12.44 | 19.33 | 55.56 |
| Shi *et al.* [17] | 48.53 | 68.47 | 75.63 | 93.02 | 16.19 | 31.44 | 39.85 | 71.13 | 8.78 | 19.90 | 27.30 | 61.20 |
| L2LTR | **56.69** | **80.86** | **87.75** | **98.01** | **26.92** | **50.49** | **60.41** | **86.88** | **13.95** | **33.07** | **43.86** | **77.65** |

same FoV and report the results in Table 4. The results show that our L2LTR, benefiting from the learnable relative positional knowledge, consistently surpasses the competing methods over three experimental settings.

**Comparison in terms of code length.** To further illustrate the advantage of our method, we compare the L2LTR with state-of-the-art methods in terms of image descriptor dimension (also called code length) in Table 2. We observe that our L2LTR has an extremely short code length of 768, which is five times shorter than that of the SAFA [16], CVFT [18], and CVM-Net [8]. A shorter code length not only implies the effective information encoding capability of the L2LTR but also means that the L2LTR provides an alternative with less storage space, lower computational complexity, and shorter running time to cross-view geo-localization.

## 4.4 Ablation Study

To investigate the effectiveness of the positional embeddings and self-cross attention mechanism, we conduct ablation studies by considering three scenarios: 1) where the positional embeddings are removed, 2) where the polar transform is removed and, 3) where self-attention replaces self-cross attention.

**Positional encoding.** We first analyze the importance of the positional embeddings and report the ablation studies on the CVUSA dataset in Table 5. From the results, we can make the following observations. First, the positional encoding endows the network with the concept of position and yields consistent improvements. In particular, adding positional embeddings improves the r@1 performance of the L2LTR from 89.04% to 91.99% and brings 3.15 points

Table 5: Ablation studies of the L2LTR.

| Model | r@1 (%) | r@5 (%) | r@10 (%) | r@1% (%) |
|---|---|---|---|---|
| Polar-L2LTR | 94.05 | 98.27 | 98.99 | 99.67 |
| w/o self-cross att. | 93.26 | 97.91 | 98.78 | 99.68 |
| w/o positional emb. | 90.90 | 97.48 | 98.40 | 99.62 |
| L2LTR | 91.99 | 97.68 | 98.65 | 99.75 |
| w/o positional emb. | 89.04 | 96.88 | 98.44 | 99.61 |

improvement at r@1 to the Polar-L2LTR. Second, we could also find that the positional embeddings are complementary to the polar transform [16, 17]. Specifically, combining the polar transform with our L2LTR improves the r@1 accuracy from 91.99% to 94.05% (+2.06%), while adding the positional embeddings to Polar-L2LTR boosts the r@1 performance from 90.90% to 94.05% (+3.15%).

**Self-cross attention.** In Table 5, we ablate self-cross attention mechanism by replacing it with self-attention on the CVUSA dataset. We can observe from Table 5 that, without imposing an increase in model complexity, self-cross attention mechanism improves r@1 performance from 93.26% to 94.05%, which manifests the effectiveness of

Table 6: Few-shot cross-view geo-localization on the CVUSA [24]. We describe the number of training pairs of each subset and its proportion (Prop.) to the original CVUSA dataset.

| Training Pairs | Prop. | Model | r@1 (%) | r@5 (%) | r@10 (%) | r@1% (%) |
|---|---|---|---|---|---|---|
| 7,106 | 20% | Polar-L2LTR | **76.01** | **90.67** | **94.01** | **98.85** |
| | | w/o self-cross att. | 75.37 | 90.42 | 92.85 | 98.66 |
| 14,212 | 40% | Polar-L2LTR | **86.06** | **95.80** | **97.16** | 99.38 |
| | | w/o self-cross att. | 85.54 | 95.14 | 97.07 | **99.42** |
| 21,319 | 60% | Polar-L2LTR | **90.30** | **96.96** | **98.26** | **99.67** |
| | | w/o self-cross att. | 88.74 | 96.60 | 98.09 | 99.66 |
| 35,532 | 100% | Polar-L2LTR | **94.05** | **98.27** | **98.99** | 99.67 |
| | | w/o self-cross att. | 93.26 | 97.91 | 98.78 | **99.68** |

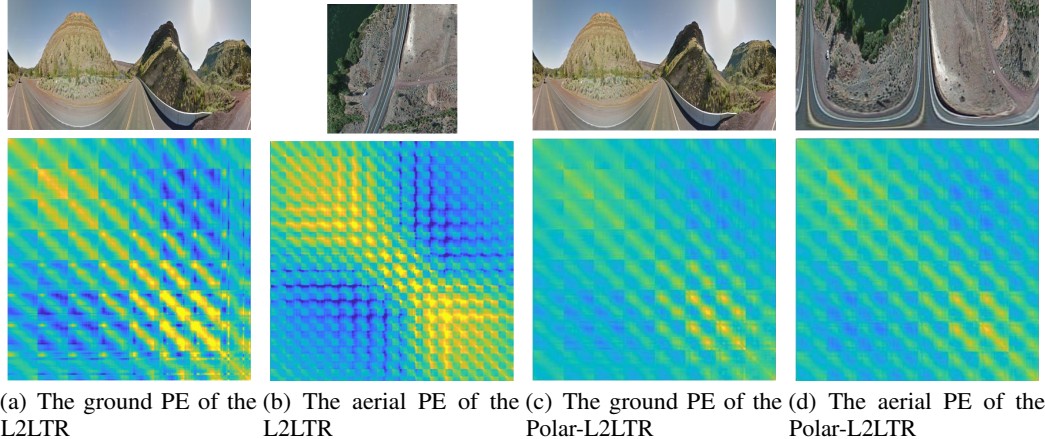

(a) The ground PE of the L2LTR    (b) The aerial PE of the L2LTR    (c) The ground PE of the Polar-L2LTR    (d) The aerial PE of the Polar-L2LTR

Figure 3: Cosine similarity between the learnable positional embeddings (PE). Yellow indicates the two positional embeddings are closer. Better viewed in color and with zoom-in.

self-cross attention. Moreover, in Table 6, we ablate self-cross attention on the few-shot cross-view geo-localization task. The few-shot task aims to learn a model that can achieve generalization from only a small number of training examples [6]. To support this task, we randomly select a certain percentage (20%/40%/60%) of samples from the CVUSA dataset to generate three subsets. The size of each subset and its corresponding proportion to the CVUSA dataset is illustrated in Table 6. Results show that replacing self-cross attention with self-attention consistently harms the network performance on the few-shot task. This indicates that self-cross attention not only improves network performance but also enhances its generalization ability. Furthermore, we also investigate how self-cross attention affects feature learning.

In Figure 4, we compare the final representation with the output of each intermediate layer by measuring their cosine similarity. As observed, replacing self-attention with our proposed self-cross attention significantly and consistently decreases the representation similarity between layers. The result implies that our self-cross attention can prevent the learned representations of Transformer layers from being overly similar, making the network more effective in capturing rich representations.

### 4.5 Qualitative Analysis

We conduct a detailed qualitative analysis on the learnable positional embeddings $\mathbf{x}_{pos} \in \mathbb{R}^{(N+1) \times D}$ to investigate whether they encode and correspond geometric configurations across views and which positional information they can learn. To this end,

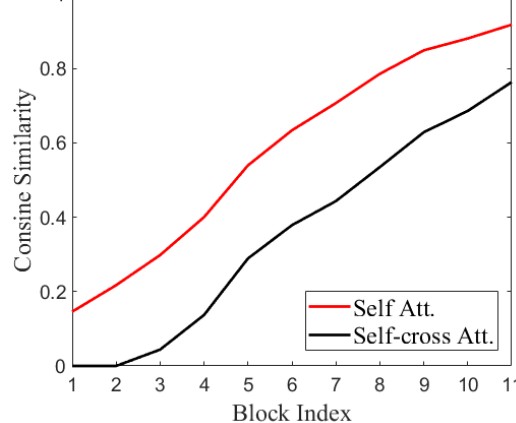

Figure 4: Cross-layer similarity between the last layer and previous layers.

we calculate the cosine similarity between two arbitrary positional embeddings of the L2LTR and acquire a distance matrix $D = Norm(\mathbf{x}_{pos})Norm(\mathbf{x}_{pos})^T$ of shape $(N + 1) \times (N + 1)$, where $Norm$ represents $L_2$ normalization. Then, we visualize the distance matrix in Figure 3, where yellow indicates the two positional embeddings are closer to each other. From Figures 3(a) and 3(b), we could find that each positional embedding is close to its neighbors with small location offsets. This implies that incorporating the learnable positional embeddings captures relative positional information. Additionally, we could observe that the visualization maps of the L2LTR are distinctly different across views in Figures 3(a) and 3(b), while the visualization maps of the Polar-L2LTR look similar to each other in Figures 3(c) and 3(d). Such similar results of the Polar-L2LTR are reasonable since the polar transform geometrically aligns cross-view images. This result further confirms that the positional encoding could capture cross-view geometric configurations.

# 5   Conclusion and Future Work

This paper proposes a novel L2LTR architecture capable of learning globally context- and position-aware representations. We also propose a novel self-cross attention to facilitate information flow across layers. Extensive experiments demonstrate that the L2LTR outperforms state-of-the-art methods in standard, fine-grained, and cross-dataset cross-view geo-localization tasks. In addition, we also conduct ablation studies and qualitative analyses to verify the effectiveness of the learnable positional embeddings and self-cross attention. One main limitation of the L2LTR is its large demand for GPU memory. Moreover, the L2LTR is built on top of the pre-trained Transformer, which requires a large amount of data for training. We aim to develop a data-efficient Transformer-based model with less memory consumption for cross-view geo-localization for future work.

## Societal Impact

This paper addresses the problem of image-based geo-localization, which benefits a wide range of applications. On the one hand, the image-based geo-localization can serve as an alternative to GPS-based localization, especially when GPS signals are jammed, blocked by buildings, or not accurate enough for specific applications. Working in conjunction with GPS-based positioning systems, our method can greatly improve the stability and safety of several downstream applications, such as autonomous driving, robot navigation, and pedestrian navigation. On the other hand, image-based geo-taggers can be used to track the location intelligence of a single image without GPS tags. This can be beneficial for applications such as digital forensics, event detection, and scene annotation. Nevertheless, negligent or malicious use of our approach could also mislead positioning systems or expose people to privacy violations. Overall, our method has both positive and negative impacts. As long as the failure cases are handled properly, and the method is not used for unethical tasks, our approach mostly leads to positive impacts.

## Acknowledgments

This work was supported by: (i) National Natural Science Foundation of China (Grant No. 62072318); (ii) Natural Science Foundation of Guangdong Province of China (Grant No. 2021A1515012014); (iii) Fundamental Research Project in the Science and Technology Plan of Shenzhen (Grant No. JCYJ20190808172007500).

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
