# OpenReview forum: "Cross-view Geo-localization with Layer-to-Layer Transformer"
_NeurIPS.cc/2021/Conference — NeurIPS 2021 Poster_

### Official Review · Reviewer_9M5m · 2021-07-08

**Rating:** 4
**Confidence:** 4

**Summary:**

This paper proposes to use transformer for cross-view image geo-localization. It adopts a cross-layer attention block to further improve the performance. Experiments on CVUSA and CVACT show improvement over existing methods.

**Limitations And Societal Impact:**

There is no discussion about societal impacts.

**Main Review:**

Strengths:
1.	The idea is simple and easy to follow.
2.	The tables are clear.
3.	The performance is higher than existing papers.

Weaknesses:
1.	Transformer has been adopted for lots of NLP and vision tasks, and it is no longer novel in this field. Although the authors made a modification on the transformer, i.e. cross-layer, it does not bring much insight in aspect of machine learning. Besides, in ablation study (table4 and 5), the self-cross attention brings limited improvement (<1%). I don’t think this should be considered as significant improvement. It seems that the main improvements over other methods come from using a naïve transformer instead of adding the proposed modification.
2.	This work only focuses on a niche task, which is more suitable for CV conference like CVPR rather than machine learning conference. The audience should be more interested in techniques that can work for general tasks, like general image retrieval.
3.	The proposed method uses AdamW with cosine lr for training, while comparing methods only use adam with fixed lr. Directly comparing with their numbers in paper is unfair. It would be better to reproduce their results using the same setting, since most of the recent methods have their code released.


**Time Spent Reviewing:**

3

---

> ### Author Response · Authors · 2021-08-10
> **REPLY TO REVIEWER 4**
>
> Please note the general comment to all reviewers regarding **model complexity** and **societal impact**.
>
> **Q1:** ***Regarding the novelty.***
>
> **A1:** Thanks for raising the concern. Our novel contributions are the proposal and evaluation of the proposed transformer-based architecture, not CNN-based, as a method for handling the correspondence problem in the cross-view geo-localization task. To our best knowledge, the Transformer has never been proposed as a cross-view geo-localization method, and the self-cross attention for Transformer has never been proposed in our research community. Below, we elaborate on our contributions and explain why these contributions are novel and non-trivial.
>
> 1. **We devise a novel Transformer-based architecture, EgoTR, which takes into account the unique characteristics of cross-view geo-localization task and serves as a highly effective method for handling the content and geometric correspondence problems in the task.** Such an attempt is novel, since we are the first to introduce the Transformer to the field of cross-view geo-localization. We note that, we are not trivially introducing the Transformer, as we make modifications in elegance to make it suit our task, instantiating a Siamese architecture with a transformer backbone to make the encoder branches domain-specific (see L152-159).  By the way, some excellent works introduce the Transformer into a specific research area, including (Lee S et al. 2021), (Zhu X et al. 2020) and (Doersch C et al. 2020). In addition, the proposed EgoTR jointly solves two key problems of existing methods, i.e. (A) disregarding global contexts and (B) imposing strong assumptions on orientation knowledge. To our knowledge, (A) has not been recognized and addressed in the cross-view geo-localization, while (B) remains an urgent yet unsolved problem in this field. Especially, addressing (B) is of great importance, “freeing the network from having to warp the aerial images through a panoramic projection” (as pointed out by R1). Also, solving (B) may also open the door to future work in geolocating orientation-unknown (as mentioned in L166-169) or planar ground images (as R1 noted), which is more common in practice.
> 2. **For the first time, we propose a self-cross attention mechanism for Transformer, adding no computational cost but improving the performance, generalization ability and training stability of network.** Such a simple yet effective modification is instrumental in improving training stability (Fig. 2 on page 5) and making the Transformer generalize better (Tab. 5 on page 8). R4 seems unsatisfied that the self-cross attention brings only 0.79% gain on the CVUSA dataset. Nevertheless, what R4 may ignore is the fact that this improvement comes with NO computational overhead. Furthermore, the self-cross attention comes with not only performance improvement, but also training stability and good generalization ability.
>
> As R4 notes, our method is simple. However, simplicity does not imply lack of novelty. Considering that the reviewers recognize that our work is novel (R1, R3) and effective (R2, R3), and that the problem we addressed is of great significance for the task (R1), we believe our research findings are worth sharing with the community.
>
> **Q2:** ***Regarding the research area.***
>
> **A2:** Thank you. We would like to clarify R4's statement as follows:
>
> 1.	Cross view geo-localization is not a niche task but a challenging task in image retrieval worth exploring in depth. In recent years, many influential works in this research area has been published in the top conferences and top journals (including NeurIPS, AAAI, CVPR, IJCV, etc.). Please refer to the references in our paper.
> 2.	Cross-view geo-localization could also benefit a wide range of downstream tasks. Please refer to [REPLY TO GENERAL ANSWER FOR ALL REVIEWERS – Q2] for more details.
> 3.	We propose a novel Transformer-based framework for the cross-view geo-localization, essentially exploring a new paradigm that is quite different from the dominant CNN-based framework.
> 4.	As suggested by R4, we will consider applying our EgoTR to general image retrieval task in future work.
>
> **Q3:** ***Regarding different training schedules.***
>
> **A3:** Our training schedule (i.e., AdamW + cosine lr) is common in Transformer-related works as it helps models converge stably. As suggested by R4, we train SAFA ([16] of the submitted paper) with AdamW optimizer instead of Adam and report the experimental results in Table VI (the result in row 1 is quoted from [16]). Note that the SAFA is the closest competitor that achieves the best and the second best results among competing methods with and without polar transform, respectively. As shown, our training strategy does slightly improve the performance of SAFA (\~0.54%). However, the SAFA is still inferior to our EgoTR, which further demonstrate the effectiveness of our method.
>
> Table VI: Comparison between our method and SAFA with the same schedule on CVUSA dataset
>
> |Training schedule|Method|Code Length|R@1|
> |:-:|:-:|:-:|:-:|
> | Adam + fixed lr |     Polar-SAFA    |   4096 |89.84%   |
> | AdamW + cosine lr schedule |     Polar-SAFA  | 4096   |   90.38%   |
> | AdamW + cosine lr schedule | Polar-EgoTR (ours) | **768** | **94.05%** |
>
>
>
> **[References]**
>
> Lee S, Yu Y, Kim G, et al. Parameter efficient multimodal transformers for video representation learning[C]// ICLR. 2021.
>
> Zhu X, Su W, Lu L, et al. Deformable DETR: Deformable Transformers for End-to-End Object Detection[C]//ICLR (oral). 2020.
>
> Doersch C, Gupta A, Zisserman A. Crosstransformers: spatially-aware few-shot transfer[C]//NeurIPS. 2020.

---

> > ### Comment · Reviewer_9M5m · 2021-08-25
> > **Response to authors**
> >
> > Thank you for the clarificaiton. It partly addresses my concerns, but some concerns still hold. After reading other reviews, I may change my rating to 5. But overall, I think the technical contribution and novelty are below the acceptance bar of NeurIPS. I agree with Reviewer 2's comments about the novelty and missing concrete explanation.
> > 1. As mentioned in the rebuttal, there are lots of works adopting transformers in visions tasks, and they have been proved to be effective. Adopting transformer for a specific task with existing components (transformer block from ViT) does not have much techinical or theoretic contribution and novelty. The only novel component here is the self-cross attention. ​However, the self-cross attention block does not seem to bring convincing improvement. Given that CVUSA is relatively small compared with ImageNet, the variance between different runs could also be 0.4%. Improvement around 0.6% is not very convincing.
> > 2. This should be fine, if AC and other reviewer are ok with this.
> > 3. The clarification addresses my concern 3, I think the authors should include the results in the paper or at least in supplementary materials.

---

> > > ### Author Response · Authors · 2021-08-31
> > > **Regarding the novelty**
> > >
> > > Thank you for engaging with our response and increasing your score. We will incorporate the experiment of Q3-A3 in the final version and address your main concern about the novelty below.
> > >
> > > ***Regarding the Transformer***
> > >
> > > We believe devising a Transformer-based architecture for the cross-view geo-localization is novel (as recognized by R1&R3) for the following reasons. Instead of trivially introducing the Transformer, to suit this task, we instantiate a Siamese architecture to make encoder branches domain-specific. Moreover, we identify the ways in which the Transformer is useful for establishing geometric and appearance correspondences across views (L69-75 of the submitted manuscript & Q1-A1). Specifically, incorporating positional knowledge helps to establish the geometric correspondence (L312-336 & L367-382 of the submitted manuscript), and we are the first to treat the acquisition of positional knowledge as a learnable problem, by attaching learnable positional embeddings to the Transformer-based baseline. This is an extremely exciting and novel development in this field, largely doing away with the need of imposing restrictive assumptions on position as in the SOTA methods ([9, 16] of the submitted manuscript). In addition, the use of global dependencies to reduce visual ambiguities in cross-view geo-localization is novel, as it has never been realized in previous works.
> > > Overall, our work is worth sharing with the community, not only for the surprising success of Transformer-based baseline over well-designed CNN-based models (see Tables 1&2), but also for the novelty as elaborated above.
> > >
> > > ***Regarding the self-cross attention***
> > >
> > > We are pleased that all the reviewers (including R4) recognized the novelty of self-cross attention. In fact, as we mentioned in R2-Q2-A2, we have included an explanation about the self-cross attention in the original manuscript. R4’s main concern is the effectiveness of this approach, which however was recognized by R2 & R3.
> > > The self-cross is effective because the advantage of self-cross attention is not only in the performance improvement, but also in improving the training stability (Fig.2 of the submitted manuscript) and the generalization ability (Tab. 5 of the submitted manuscript). Since Transformers notoriously suffer from the instability of training, we hope that our proposed method could shed light on designing stable Transformers. Also, the superiority of self-cross attention on few-shot geo-localization suggests a broader practical applicability in real-world scenarios where data is often scarce. In addition, this improvement comes with NO computational overhead, which further demonstrates the superiority of self-cross attention.

---

> > > > ### Comment · Area_Chair_8g9B · 2021-09-01
> > > > **Score updated to 5**
> > > >
> > > > Dear Reviewer 9M5m,
> > > >
> > > > After carefully reading the paper and other reviews, and reading your own comment ("After reading other reviews, I may change my rating to 5") I am considering your score to be 5, not 4. I have also dismissed claims of “novelty” and “narrowness of scope”.
> > > >
> > > > Best wishes,
> > > > AC

---

### Official Review · Reviewer_Bnto · 2021-07-12

**Rating:** 6
**Confidence:** 5

**Summary:**

a) This paper brings transformer to solve the cross-view geo-localization problem;
b) The main contributions are two folds: one using transformer and the other using a modified attention scheme;
c) Results are good and outperform previous method.



**Ethics Review Area:**

["I don’t know"]

**Limitations And Societal Impact:**

Yes.

**Main Review:**

Strong points:

a) The paper is well-written;

b) Using transformer for cross-view geo-localization is novel;

c) Though modifying the attention equation is engineered, it is effective for this task;

d) Experimental results are good.


Weak points:

a) I would say the claim of the superiority of learned position embeddings is too strong and expect the following experiments.

Randomly orientate the ground-view images and compare the proposed method with respect to [17].

Without this experiment, the proposed method can also benefit from the regularized configuration of aligned cross-view images.


b)  Self-cross attention.

x_{l-1} and x_{l-2} is used  in equation (3), can we use x_{l-3} to replace x_{l-2}?

or, jointly using x_{l-1}, x_{l-2} and x_{l-3}?

x_{l-3} can be used to build to Q_l.


c) Time and model complexity comparisons.

Though the code length [line 300] is short for the proposed method, as the transformer is time-consuming, I would expect detailed time and model complexity comparisons with respect to previous methods.




**Time Spent Reviewing:**

3 hours

---

> ### Author Response · Authors · 2021-08-10
> **REPLY TO REVIEWER 3**
>
> Please note the general comment to all reviewers regarding **model complexity** and **societal impact**.
>
> **Q1:** ***Regarding the effectiveness of the proposed method on randomly orientated ground-view images.***
>
> **A1:** In the submitted paper, we focus mainly on matching panoramas to aerial images, which has the same setting as the SOTA methods and provides more information for learning cross-view correspondence. As R3 suggested, we conduct experiments on planar ground images that have unknown orientation and limited field of view (FoV=180°) as in [17] (the submitted paper) to further demonstrate the superiority of EgoTR. We follow the same experimental setting as [17] and report the experimental results in Table IV. It can be observed that our EgoTR outperforms the competing method in geolocating planar ground images, which demonstrates that our method is more reliable in geolocating planar ground images.
>
> Table IV: Comparison between EgoTR and [17] on localizing planar ground images
>
> |Method|R@1|R@5|R@10|R@1%|
> |:-:|:-:|:-:|:-:|:-:|
> |Shi et al. [17]|48.53%|68.47%|75.63%|93.02%|
> |Ours|**56.69%**|**80.86%**|**87.75%**|**98.01%**|
>
> **Q2:** ***Regarding different inputs of the self-cross attention.***
>
> **A2:** Considering the semantic gap between feature maps caused by different network depths, we do not use all intermediate features for the self-cross attention, but only the highly correlated adjacent features (see L57-59). To better illustrate this point, we conduct experiments by replacing $x_{l-2}$ with $x_{l-3}$ in Eq. 3 (the submitted paper) and report the experimental results on the CVUSA. As can be clearly observed below, using the $x_{l-3}$ degrades the R@1 performance by \~ 7 points, which further reinforces our previous analysis.
>
>
> Table V Comparison of accuracy in different self-cross attention
>
> |Method|R@1|R@5|
> |:-:|:-:|:-:|
> |Using $x_{l-1}$ and $x_{l-2}$|**94.05%**|**98.27%**|
> |Using $x_{l-1}$ and $x_{l-3}$|87.44%|96.56%|
>
> **Q3:** ***Regarding the model complexity.***
>
> **A3:** Please refer to [REPLY TO GENERAL ANSWER FOR ALL REVIEWERS – Q1] for more details.

---

### Official Review · Reviewer_JyiQ · 2021-07-17

**Rating:** 6
**Confidence:** 5

**Summary:**

This paper proposes EgoTR for cross-view geo-localization with evolving transformer. Specifically, it utilizes the properties of self-attention in transformers to model global dependencies, which reduces visual ambiguities in cross-view geo-localization. The positional encoding is also used in the proposed EgoTR to understand and correspond geometric configurations between ground and aerial images.

**Ethical Concerns:**

none.

**Limitations And Societal Impact:**

Limitations are discussed in conclusion, but not societal impact.

**Main Review:**

- This paper is well-written and easy to follow.

- The paper claims that the use of positional encoder removes the need for strong assumptions on geometry knowledge required in some existing cross-view geo-localization methods. It is however not very obvious from the paper how this is achieved over the cross-view setting. It seems like the positional encoding mentioned in Line 160-171 is based on the image patches in the respective domains. Thus, shouldn't the positional knowledge be just within the image in the same domain, i.e. self-attention? How would the Siamese network architecture and the Triplet loss help in enhancing the cross-view geometry knowledge via the positional encoder?

- The individual components of the proposed framework is not new. Transformer is from ViT [4], and the loss function is from [8]. The only new thing is the attention across different layers in the network proposed in Section 3.3. Therefore, this paper can be seen as another additional paper that introduces transformer into a well-established task / framework.

- What is the explanation on self-attention across different layers can help cross-view localization? Similar to the positional encoding, it is also done within the same view. The only interaction across the two views is the final feature embeddings and the triplet loss.

- Using transformers mean that the memory consumption might increase a lot compared to other existing methods.

- The results on the two benchmark datasets show good performances on the proposed method. This seems to empirically suggest that the method that adopted transformer is effective on the task of cross-view geo-localization.

- The ablation studies in Section 4.4 also seem to empirically support that claim that the positional encoder and self-cross attention across different layers are effective. However, as mentioned earlier, a more concrete explanation on why positional encoder and self-attention across layers can help cross-view geo-localization is missing.



**Time Spent Reviewing:**

2 hours

---

> ### Author Response · Authors · 2021-08-10
> **REPLY TO REVIEWER 2**
>
> Please note the general comment to all reviewers regarding **model complexity** and **societal impact**.
>
> **Q1:** ***Regarding why the positional embeddings can help cross-view geo-localization.***
>
> **A1:** As we highlight in Section 4.5, the learnable positional embeddings encode positional information specific to the ground or aerial domain (as R2 notes). As pointed out by (Liu & Li. 2019), adding domain-specific position maps to features can teach and endow with a Siamese network the notion of position. When cross-view features and their associated positional embeddings are jointly fed to the Siamese-like Transformer for projecting cross-domain features to a common feature space, such geometric knowledge can be effective in avoiding the ambiguities caused by geometry misalignments across views. And the triplet loss, which brings matching pairs closer while pushing non-matching pairs far apart, is used as an objective function to help reduce the cross-domain discrepancy and project cross-domain features into a common space. We have included the above details into the revised version for a better understanding.
>
> **Q2:** ***Regarding why the self-cross attention can help cross-view geo-localization.***
>
> **A2:** In the submitted paper, we do investigate and explain how the self-cross attention helps to improve the network performance. Specifically, as we have stated in Section 3.3 (L180-184), the proposed self-cross attention creates short path between adjacent layers, thus allowing information flows effectively across layers. Then, we empirically show how this would affect feature learning. First, in Fig. 2, we show that, the self-cross attention leads to more stable network training. Since Transformers notoriously suffer from the instability of training, we hope that our proposed self-cross attention could shed light on designing stable Transformers. Second, in Fig. 3, we find that, the short path between adjacent layer could also prevent Transformer layers from producing overly similar intermediate features and from being less effective for capturing rich representations. Because the key to cross-view geo-localization is to learn a powerful representation, we show in Section 4.4 (Tables 4&5) that the self-cross attention does improve the network’s performance and generalization ability for the cross-view geo-localization.
>
> **Q3:** ***Regarding introducing transformer into a well-established task/framework.***
>
> **A3:** Thanks for raising the concern. Besides the self-cross attention, our another contribution is the proposed transformer-based architecture, not CNN-based, as a method for handling the correspondence problems in the cross-view geo-localization task. To our best knowledge, the Transformer has never been proposed as a cross-view geo-localization method, and the self-cross attention for Transformer has never been proposed in our research community. Below, we elaborate on our contributions and explain why these contributions are novel and non-trivial.
>
> 1) **We devise a novel Transformer-based architecture, EgoTR, which takes into account the unique characteristics of cross-view geo-localization task and serves as a highly effective method for handling the content and geometric correspondence problems in the task.** Such an attempt is novel, since we are the first to introduce the Transformer to the field of cross-view geo-localization. We note that, we are not trivially introducing the Transformer, as we make modifications in elegance to make it suit our task, instantiating a Siamese architecture with a transformer backbone to make the encoder branches domain-specific (see L152-159). In addition, the proposed EgoTR jointly solves two key problems of existing methods, i.e. (A) disregarding global contexts and (B) imposing strong assumptions on orientation knowledge. To our knowledge, (A) has not been recognized and addressed in the cross-view geo-localization, while (B) remains an urgent yet unsolved problem in this field. Especially, addressing (B) is of great importance, “freeing the network from having to warp the aerial images through a panoramic projection” (as pointed out by R1). Also, solving (B) may also open the door to future work in geolocating orientation-unknown (as mentioned in L166-169) or planar ground images (as R1 noted), which is more common in practice.
> 2) **For the first time, we propose a self-cross attention mechanism for Transformer, adding no computational cost but improving the performance, generalization ability and training stability of network.** Such a simple yet effective modification is instrumental in improving training stability (Fig. 2 on page 5) and making the Transformer generalize better (Tab. 5 on page 8). We highlight that the performance improvement of self-cross attention comes with NO computational overhead, and that the self-cross attention comes with not only performance improvement, but also training stability and good generalization ability.
>
> Considering that the reviewers recognize that our work is novel (R1, R3), simple (R4) and effective (R2, R3), and that the problem we addressed is of great significance for the task (R1), we believe that our research findings are worth sharing with the community.
>
> **Q4:** ***Regarding the memory consumption.***
>
> **A4:** Although the Transformer-based model increases the memory consumption, our EgoTR achieves the best performance compared to the CNN-based methods. Please refer to [REPLY TO GENERAL ANSWER FOR ALL REVIEWERS – Q1] for more details.
>
> **[Reference]**
>
> Liu L, Li H. Lending orientation to neural networks for cross-view geo-localization[C]//CVPR. 2019.

---

### Official Review · Reviewer_kVJu · 2021-07-26

**Rating:** 7
**Confidence:** 4

**Summary:**

The paper tackles the cross-view geo-localization problem, which seeks to match a given ground-level image to aerial images that cover the same area, selected from a fixed database.

The paper is the first to apply transformers to the problem, and reports state-of-the-art results. Notably, the transformer’s attention mechanism largely frees them from having to warp the aerial image to match the ground-level image’s panoramic projection, a procedure that can go awry in the face of significant occlusion in the ground-level panorama, and/or errors in the ground pano’s orientation data. The resulting accuracy improves the state of the art by a significant margin.

The paper also makes a few adjustments to the ViT transformer, performing ablation studies on some of them.

**Ethical Concerns:**

In the checklist, item 1c (“did you discuss any potential negative societal impacts of your work?”), the answer provided is [N/A].

I can easily imagine one negative societal impact: people who have taken 360-degree photos (e.g. GoPro Max, Insta360 One X2) with geo-tagging deliberately turned off are now at risk of being geo-tagged. As the field matures, conventional planar photos may also be geotagged by the pixels alone.

On the positive end, such geotaggers could help Bellingcat-style digital forensics.

**Limitations And Societal Impact:**

In section 5, the authors highlight that the main limitation of EgoTR is its large demand on GPU memory. Please quantify this demand and compare it to that of the closest competitors.

Line 391-392 says “Moreover, the EgoTR is built on top of the pretrained Transformer, which requires a large amount of data for training.” Is it the pre-training or the fine-tuning that takes a lot of data? Please quantify the data demands and compare it to that of the closest competitors, for example by comparing the number of epochs it takes to converge.

Can you foresee any limitations that would impede success when using this method to match planar aerial images to planar ground images?

**Main Review:**

# Originality
The paper is the first to apply the transformer attention architecture to this problem. In addition, it modifies the ViT (Vision Transformer) [4] in a few ways: it replaces image patches with single-pixel columns through the feature maps, which strikes me as an improvement in elegance. It also computes the keys of layer i from the values of layer i-1. Finally, it introduces a ResNet-like pass-through connection through the transformer stack. The paper presents ablation studies of the effects of the first two changes above.

# Quality
The submission is technically sound, and analyses presented show good evidence of its accuracy advantage over its predecessors. The ablation studies are useful and well presented.

There are some claims that are unsupported by evidence. Notably, the explanation in lines 180-198, of how self-cross attention affects feature learning, seems weak.

Lines 181-185 mention an architectural similarity to ResNet. In the paragraph below labeled “Clarity”, I talk about how the wiring of Fig 1b is messy. As it stands, the figure obscures any parallel with ResNet. Once fixed, this parallel should be more visually obvious.

In lines 189-191: “As shown, during the early training stage, the localization performance of the self-attention-based model fluctuates a lot.” It isn’t obvious that the early stage fluctuations of the red curve are significantly worse than those of the black curve. Could you quantify this, e.g. by computing a variance envelope over a moving time window? In any case, the final performance gap between the red and black curves could be attributable to many many things, and it’s not at all obvious why the early stability advantage of the black training curve (assuming there is one)  should be singled out as an explanation. Please remove the statement “As a result, the stable training of the EgoTR could improve the network performance.” Without direct evidence, this is purely speculative. In any case, even if the early stability did lead to better final performance, this only replaces one question (“what causes the better performance”) with a speculative statement (“stable training causes better performance”) tacked onto another question (“what causes the stability?”). That question is just as open as the original question.

“Compared to the self-attention in Eq. 2, our proposed self-cross attention creates short path between adjacent layers,”

# Clarity
There are a number of places where the clarity could be improved:

## Abstract

“Evolving”:
I found the paper’s use of the term “Evolving” to be confusing, in the sense that it led me astray from what is actually being described. In a field where evolution mechanisms are closely studied as a part of the learning toolbox, I would stay away from using “evolve” to describe anything that doesn’t involve Darwinian (or even Lamarckian!) improvement. Even when taking the more relaxed view that “evolve” can refer to any sequential improvement process, it seems misapplied to the EvoTR. While the keys of layer i are a function of the features of layer i-1, It’s hard to say that the keys are “improved” from one layer to the next. Each set of keys are specific to that layer’s features, and comparing them across layers would be comparing apples to oranges. In that case, what is being “evolved”, i.e. sequentially improved over iterations?

I recommend renaming EvoTR to replace “evolve” with something closer to what is being described, for example, L2LT (Layer-to-layer Transformer).

## Section 3.2
The wiring in Fig. 1b is messy. At the bottom, x_{i-1} goes into Layer Norm, the output of which should be z_l according to Eq 3. This z_l is unnecessarily merged with the wire labeled x_{l-2} (did you mean for that to be z_{l-1}?). These wires should be kept separate: the z_l wire should go to the Q and V blocks, while the z_{l-1} wire should go to the K block. Most importantly, under the present figure it’s unclear where the x_{i-1} output at the top of the figure comes from. It can’t be the output of the Layer Norm; that’s z_l. It’s not the x_{l-2} input at the bottom; the index is different. Please fix the labels if necessary, and when two wires merge, make sure they only do so by going into a labeled block (e.g. “Add”).

In “Preliminaries: Vision Transformer”, please explain what “class” means, in the context of the cross-view geo-localization problem. Most readers will be familiar with “class” as used in image classification, so it is unusual to see “class” be part of the input data, as opposed to being a target output.

As you describe the ViT here, please be explicit about which parts correspond to the different dashed boxes in Fig 1a (even understanding that Fig 1a contains a modified ViT, rather than the vanilla ViT being described here).

In “Learnable positional embedding”, please describe why the positional embedding is added to the feature maps, rather than concatenated as additional channels. It’s a strikingly surprising design choice, at least to me.

Please separate Equations 2 and 3 into separate numbered equations. If you must keep the equations for Q_l, K_l, and V_l on a single line, please add more whitespace after the commas, to keep them visually separate.

## Section 4.1
Please provide more details on the difference between CVACT vs CVACT_test. It sounds like the former provides just pairs of ground images with their corresponding aerial images. Is this correct? CVACT sounds the same, except that both images now have “geo-tags”, which I assume are like GPS coordinates? Is this correct? If so, How is this extra information used, during inference, training, and evaluation? Also, why is the test set (CVACT_test) ~3x the size of the CVACT training set?

## Section 4.2
Consider moving section 4.2 (“implementation detail”) to the beginning of section 4 (as 4.1).

Please provide more detail on the pretraining. In particular:
* Did you train a single encoder to classify ImageNet images, then transfer those same trained weights to both the aerial and ground encoder networks?
* Did you augment ImageNet images with positional information?
* How does the network do without pre-training?

## Page 8
Most of the page makes reference to Table 4. Please try to move that figure to this page.

## Section 4.5
Please go into more detail on how to interpret the distance matrices of figure 4. In distance image D, is D[i, j] the dot product of the positional embeddings of pixel V[i] and pixel V[j], where V is the flattened version of the ground/aerial image? Where does the checkerboard pattern come from? Without having this nailed down, it’s hard to evaluate the claims being made in this section, especially the one in lines 375-378. Lines 379-382 need an image illustrating the effect they describe, as it’s hard to visualize.

## Nit
Throughout the paper, there are misused articles, mostly in the form of unnecessary or missing “the'''s. Before submission, please run the paper by a colleague who speaks English natively, or if none are available, through a grammar checker (e.g. Grammarly).

# Significance
The results are significant. The architecture presented is an improvement in accuracy over competing methods. They largely do away with the need to preprocess the aerial image through a panoramic projection, a potentially brittle step that we are well rid of. The claim is that the transformer’s attention mechanism handles the correspondence problem instead. If this is true, this work may also open the door to future work in geolocating planar ground-level images, which are far more common than ground panoramas. Unfortunately, this is not studied or mentioned in this paper.


**Time Spent Reviewing:**

6

---

> ### Author Response · Authors · 2021-08-10
> **REPLY TO REVIEWER 1 (1/2)**
>
> Please note the general comment to all reviewers regarding **model complexity** and **societal impact**.
>
> **Q1:** ***Regarding Fig 1.***
>
> **A1:** Thank you. We have made the following revisions in the revised version.
>
> 1) At the bottom of Fig. 1b, we separate $z_{l}$ and $z_{l-1}$ wires: the $z_l$ wire goes to the Q and V blocks, while the $z_{l-1}$ wire goes to the K block.
> 2) We feed $x_{l-2}$ into the “Layer Norm” block to acquire $z_{l-1}$ as the K block’s input, instead of feeding the $x_{l-2}$ directly into the K block.
> 3) The $x_{l-1}$ output at the top of Fig. 1b is exactly the input $x_{l-1}$ (at the bottom of Fig. 1b), not the $z_l$ or the $x_{l-2}$. We have corrected this error.
> 4) We highlight the short path between adjacent layers in Fig. 1b to emphasize the parallel with ResNet.
> 5) In the revised version, we have explicitly indicated in the paper the parts corresponding to the dashed boxes in Fig. 1a.
>
> **Q2:** ***Regarding quantifying the statement of L189-191 by a variance envelope.***
>
> **A2:** We thank R1 for her/his constructive suggestion. This will help to clarify more clearly the effect of self-cross attention. We have included a quantitative comparison into Fig. 2 by adding a variance envelope in the revised version.
>
> **Q3:** ***Regarding removing the statement "stable training improves the performance".***
>
> **A3:** We will delete the statement "the stable training of the EgoTR could improve the network performance", which is not rigorous enough. In the revised version, we only highlight that the self-cross attention leads to more stable early training, without directly correlating stable training with better performance. Since Transformers notoriously suffer from the instability of training, we hope that our proposed self-cross attention could shed light on designing stable Transformers.
>
> **Q4:** ***Regarding the term “Evolving”.***
>
> **A4:** In our experiments (L357-363), we compare the final features with the intermediate features of each encoder block of EgoTR, showing that the self-cross attention could prevent the learned representations of the Transformer blocks from being overly similar to each other. For a more concise presentation, we adopt the term “evolve” to describe this result and name our network as "Evolving Transformer".  R1’s statement on “evolving” is enlightening. As R1 suggested, Layer-to-layer Transformer could be more appropriate. We have considered this suggestion throughout our revised manuscript very carefully.
>
> **Q5:** ***Regarding the term "class" in “Preliminaries: Vision Transformer”.***
>
> **A5:** In several Transformer-related works, such as BERT (Devlin J et al. 2019) and [4] (the submitted paper), it is common to prepend a class embedding to the input sequence, and the final hidden state corresponding to the class embedding serves as the aggregate sequence representation for the classification task. In “Preliminaries: Vision Transformer”, we follow the naming convention in [4] and denote the prepended learnable embedding of the input as a “class embedding”. The main difference is that the aggregate representations we obtained are not used for the classification task, but for the cross-view geo-localization task.
>
> **Q6:** ***Regarding adding rather than concatenating the positional embeddings.***
>
> **A6:** For clarity, in the following we reuse the notation of L140-145 (the submitted paper).
>
> There are two main reasons why we choose to add the positional embeddings to the feature maps. First, obviously, concatenating the positional embeddings $x_{pos}$ as additional channels comes at an additional computational cost, increasing the computational complexity of self-cross attention and FFN. Second, concatenating the positional embeddings as additional channels disentangles the semantic and positional information. Yet, the spatial positions of scene images are highly related to their semantics (Choi S et al. 2020). For example, sky and road mainly lie in the upper and lower parts of a ground image, respectively. This implies that the semantics may help to effectively learn positional information when using the element-wise addition. As a result, considering the unique characteristic of cross-view geo-localization, such a common concatenation practice may not be appropriate. The experimental results shown in Table II reinforce our analysis.
>
> Table II: Comparison of adding and concatenating the positional embeddings
>
> | |R@1|R@5|R@10|R@1%|
> |:-:|:-:|:-:|:-:|:-:|
> |Add|**94.05%**|**98.27%**|**98.99%**|**99.67%**|
> |Concatenate|88.06%|96.68%|97.85%|99.48%|
>
> **Q7:** ***Regarding the difference between CVACT vs CVACT_test.***
>
> **A7:** First, CVACT dataset contains 35,532/8,884/92,802 image pairs for training/validation/test and is accompanied with UTM coordinates (i.e., the geo-tags). We denote the CVACT training/validation/test sets as CVACT_train/val/test, respectively. Second, the CVACT_test differs from the CVACT_train and CVACT_val in its retrieval setting. Specifically, for the CVACT_train and CVACT_val, the correct match of a ground image is the corresponding aerial image (one-to-one retrieval), while for the CVACT_test, a retrieved aerial image is considered correct as long as it is within the distance d=5m (computed by the UTM coordinates) from the ground-truth location of the ground image. That is, in the CVACT_test, there might be multiple reference aerial images for a given ground image, which is a more realistic and practical setting. In the submitted paper, we denote the tasks performed on the CVACT_val and the CVACT_test as “standard” (L246) and “fine-grained” (L265) cross-view geo-localization tasks, respectively. Third, the CVACT_test is a large-scale test set with 92,802 image pairs densely covering a city. This allows for not only the fine-grained geo-localization but also a thorough investigation of the generalization ability of our EgoTR. We have included more details about the datasets in the revised version.
>
> **Q8:** ***Regarding the pre-training.***
>
> **A8:** 1) Yes. We train a single encoder to classify ImageNet images and transfer the same weights to both the aerial and ground encoder networks. 2) Yes. During pre-training, we attach positional embeddings to ImageNet images. 3) In line with most of existing works [2,9,16,17,18] (the submitted paper), we use a pre-trained model and fine-tune it on CVUSA or CVACT datasets. In Table III, we show the experimental results of our network without pre-training, observing a performance degradation on CVUSA dataset.
>
> Table III: Model performance with and without pre-training
>
> |Pre-training|R@1|
> |:-:|:-:|
> |Without|75.83%|
> |With|**94.05%**|
>
>
> **Q9:** ***Regarding the distance matrices of figure 4 and the checkerboard pattern.***
>
> **A9:** For more clarity, in the following we reuse the notation of L140-145 (the submitted paper).
> First, for the given positional embeddings $x_{pos}$ of shape $(N+1)\times D$, where $N$ is the number of patches, and $D$ is the embedding size, we compute the dot product between two arbitrary L2 normalized positional embeddings and obtain a distance matrix of shape $(N+1)\times(N+1)$. Note that $N$ is set to 256, and $D$ is set to 768.
> Second, to interpret the checkerboard pattern, it is critical to understand what information is encoded in each position vector. For a given ground feature map extracted from the backbone with a spatial size of $8\times 32$, we flatten it to $8\times32=256$ tokens as the input of encoder (denoted as $x$). After adding the $x_{pos}$ to the $x$ (we ignore $x_{class}$ here for simplicity), the $x_{pos}[(k-1)\times 32: k\times 32]$ corresponds to the positional information of the $k$-th row in the ground feature map ($k \in{1,…,8}$). Then, it's easy to understand why the checkerboard pattern exists. Take the $x_{pos}[0]$ as an example. Due to the 2D nature of an image, the $x_{pos}[0]$ should have a high similarity not only with the $x_{pos}[1]$ but also with the $x_{pos}[32]$. This phenomenon is reflected in the image by the high response value in the top left corner of the checkerboard in row 1 and column 2. By analogy, for the $x_{pos}[0]$, the high response values appear every 32 pixels in the distance map, so that there will be 8 checkerboards blocks in the distance image, which is consistent with Fig. 4a. Similar observations hold for the satellite distance image.
> Third, for a better understanding, we have incorporated an image illustrating the concept introduced in L379-382.

---

> > ### Author Response · Authors · 2021-08-10
> > **REPLY TO REVIEWER 1 (2/2)**
> >
> > **Q10:** ***Regarding the effectiveness of the proposed method on planar ground-level images.***
> >
> > **A10:** We are encouraged that R1 recognizes the significance of our proposed architecture. In the submitted paper, we focus mainly on matching panoramas to aerial images, which has the same setting as the SOTA methods and provides more information for learning the cross-view correspondence. As R1 suggested, we conduct experiments on planar ground images that have unknown orientation and limited field of view (FoV=180°) to further demonstrate the superiority of EgoTR. Specifically, we randomly shift and crop the ground panoramas of the CVUSA dataset along the horizontal direction to generate the planar ground images. We train on the cropped CVUSA dataset and report the experimental results in Table IV. Note that due to time constraints, we only compare with [17] (the submitted paper). It can be observed that our EgoTR outperforms the competing method in geolocating planar ground images, which demonstrates that our method is more reliable in geolocating planar ground images.
> >
> > Table IV: Comparison between EgoTR and [17] on localizing planar ground images
> >
> > |Method|R@1|R@5|R@10|R@1%|
> > |:-:|:-:|:-:|:-:|:-:|
> > |Shi et al. [17]|48.53%|68.47%|75.63%|93.02%|
> > |Ours|**56.69%**|**80.86%**|**87.75%**|**98.01%**|
> >
> >
> >
> > **Q11:** ***Regarding the data demands of pre-training and fine-tuning.***
> >
> > **A11:** Our EgoTR requires more data for pre-training because the Transformer lacks image-specific inductive biases, which is inherent to CNNs. However, the pre-training process can be efficiently parallelized across distributed GPUs, thus significantly reducing the pre-training time. For fine-tuning, the EgoTR requires the same amount of data (35,532 image pairs for CVUSA or CVACT datasets) as SOTA methods. Moreover, during training phase, our EgoTR converges in \~11K steps, while SAFA ([16] of the submitted paper) converges in \~30K steps (\~19K more steps than ours).
> >
> > **Q12:** ***Regarding the limitations of using this method to locate planar ground images.***
> >
> > **A12**: As shown in Table IV, our EgoTR achieves impressive performance in matching planar ground images to aerial images. Below we show several limitations that can hinder success when geolocating planar ground images. First, a single planar ground image contains limited information from its limited field of views. In some extreme cases, such as facing a large area of sea, our method will have difficulty in geolocating the planar ground images. Second, as most of the ground images are taken during the daytime hours for training, various conditions, such as night, different seasons and weather, may affect the accuracy of our method in practical applications.
> >
> > **Q13:** ***Regarding minor points.***
> >
> > **A13:** Thank you for your careful reading and good suggestions. We have carefully modified the indicated issues one by one and polished the writing of the paper in the revised revision.
> >
> > **[References]**
> >
> > Choi S, Kim J T, Choo J. Cars can't fly up in the sky: Improving urban-scene segmentation via height-driven attention networks[C]//CVPR. 2020.
> >
> > Devlin J, Chang M W, Lee K, et al. BERT: Pre-training of Deep Bidirectional Transformers for Language Understanding[C]//NAACL-HLT (1). 2019.
> >
> > Dosovitskiy A, Beyer L, Kolesnikov A, et al. An Image is Worth 16x16 Words: Transformers for Image Recognition at Scale[C]//ICLR. 2020.

---

> > > ### Comment · Reviewer_kVJu · 2021-09-10
> > > **Thorough**
> > >
> > > Many thanks for thoroughly addressing the comments and adding detail where requested. I am reassured in maintaining my recommendation to accept.

---

### Author Response · Authors · 2021-08-10
**General Answer for all Reviewers**

Heartfelt thanks go to all the reviewers for their time and efforts spent on our paper. We are encouraged that the reviewers found our ideas novel (R1, R3), simple (R4) and effective (R3), the issues addressed in this paper significant (R1), our experimental results impressive (R1, R2, R3, R4), and our paper well-written and easy to follow (R2, R3).

We address the common comment on model complexity and societal impact below.

**Q1:** ***Regarding the model complexity.***

**A1:** In section 5, we highlight the complexity limitation we would like to address in future work. Here, we quantify such limitation and compare our EgoTR to SAFA ([16] of the submitted paper), the closest competitor that achieves the best and the second best results among competing methods with and without polar transform, respectively. Note that the following results are obtained without applying the polar transform, which is a more realistic setting for real-world applications.

Table I: Complexity comparison on CVUSA dataset

| Model | GFLOPs | Time Complexity | Inference Time | Retrieval Time |Code Length | R@1|
|  :----:  | :----:  |:----:  |:----:  |:----:  |:----:  |:----:  |
| SAFA  | **42.19** | $O( M^2_l K^2_l d_{l-1} d_l )$ |**5.74ms** | 249ms |  4096 | 81.15% |
|EgoTR (ours) | 44.06 | $O(n^2 d_l)$ |6.07ms |**179ms** |  **768** | **91.99%** |

For layer $l$, $M_{l}$ and $d_l$ denote the spatial size and channels of feature maps, respectively, while $K_l$ indicates the convolutional kernel size. $n$ is the number of image patches in Transformer. Our EgoTR achieves the best performance (91.99% vs. 81.15% at R@1) on CVUSA dataset with a shorter code length (768 vs. 4096).  Although the EgoTR runs 0.33ms slower than the SAFA (\~6% more inference time spent on encoding an input), it retrieves 70ms faster than the SAFA (\~28% shorter retrieval time spent on returning the best match).

**Q2:** ***Regarding the societal impact.***

**A2:** Sorry for not discussing the social impact in detail. This paper addresses the problem of image-based geo-localization, which benefits a wide range of applications. On the one hand, the image-based geo-localization can serve as an alternative to GPS-based localization, especially when GPS signals are jammed, blocked by buildings or not accurate enough for specific applications. Working in conjunction with GPS-based positioning systems, our method can greatly improve the stability and safety of several downstream applications, such as autonomous driving, robot navigation, pedestrian navigation, etc. On the other hand, image-based geo-taggers can be used as a tool for tracking location intelligence of a single image without GPS tags, which can be beneficial for applications such as digital forensics, event detection, scene annotation, etc. Nevertheless, negligent or malicious use of our approach could also mislead positioning systems or expose people to privacy violations. Overall, our method has both positive and negative impacts. As long as the failure cases are handled properly and the method is not used for unethical tasks, our approach mostly leads to positive impacts. The above discussion has been added to the revised version.

---

### Decision · Program_Chairs · 2021-09-27

**Decision:**

Accept (Poster)

**Comment:**

This paper proposes a siamese convolutional + vision transformer architecture for cross-view geolocalisation, from ground view equirectangular panorama images to satellite views (cartesian or polar coordinates). The transformer has a cross-attention mechanism between consecutive layers (I was confused by the word “evolving”, since no genetic algorithms were involved). The method is thoroughly evaluated on a US cross-modal localisation dataset where it achieves state-of-the-art results.

Reviewer kVJu praised the idea and significance of results but criticised some speculative or unsubstantiated claims in the paper, as well as the clarity, including the usage of term “evolution”; the authors responded with a long list of modifications and additional results. Reviewer JyiQ was uncertain why positional encoding and cross-layer self-attention specifically helped with the localisation task (the authors’ response was only empiriical but did not provide intuition about why it worked). Reviewer Bnto had questions about ground view image orientations, which were answered by the authors through additional experiments. Reviewer 9M5m made the usual comment about novelty (which I will dismiss) and narrowness of scope (again, that’s an easy critique to make), as well as a claim about learning rate (addressed by the authors).

All scores but for reviewer 9M5m (4) were positive (6, 6, 7). Reviewer 9M5m said they would update their score to 5 (which makes the paper average equal to 6). I have dismissed claims of “novelty” and “narrowness of scope” and decided to accept the paper.